Multiple propane gas burn rates procedure to determine accuracy and linearity of indirect calorimetry systems: an experimental assessment of a method

Ismail Mohammad 1
Alsubheen Sanaa A. 2
Loucks-Atkinson Angela 1
Atkinson Matthew 3
Alkanani Thamir 1
Kelly Liam P. lpkelly@mun.ca 4
Basset Fabien fbasset@mun.ca 1
1 School of Human Kinetics and Recreation, Memorial University of Newfoundland , St John’s , Newfoundland , Canada
2 School of Physical Therapy Faculty of Health and Rehabilitation Sciences, Western University , London , Ontario , Canada
3 Department of Mathematics and Statistics, Memorial University of Newfoundland , St. John’s , Newfoundland , Canada
4 Faculty of Medicine, Memorial University of Newfoundland , St. John’s , Newfoundland , Canada
Urban Pawel
Electronic publication date: 2022 Aug 29
Publication date: 2022
Volume: 10
Electronic Location ID: e13882
Received 2019 Feb 26; Accepted 2022 Jul 20
Copyright: ©2022 Ismail et al.
Copyright year: 2022
Copyright holder: Ismail et al.
License: This is an open access article distributed under the terms of the Creative Commons Attribution License, which permits unrestricted use, distribution, reproduction and adaptation in any medium and for any purpose provided that it is properly attributed. For attribution, the original author(s), title, publication source (PeerJ) and either DOI or URL of the article must be cited.
License URL: https://creativecommons.org/licenses/by/4.0/

Keywords: Indirect calorimetry, Propane gas, Energy production, Accuracy, Linearity

Funding: School of Human Kinetics and Recreation This work was supported by the School of Human Kinetics and Recreation. The funders had no role in study design, data collection and analysis, decision to publish, or preparation of the manuscript.

==============================
Objective

Indirect calorimetry (IC) systems measure the fractions of expired carbon dioxide (FeCO2), and oxygen (FeO2) recorded at the mouth to estimate whole-body energy production. The fundamental principle of IC relates to the catabolism of high-energy substrates such as carbohydrates and lipids to meet the body’s energy needs through the oxidative process, which are reflected in the measured oxygen uptake rates (V̇O2) and carbon dioxide production rates (V̇CO2). Accordingly, it is important to know the accuracy and validity of V̇O2and V̇CO2 measurements when estimating energy production and substrate partitioning for research and clinical purposes. Although several techniques are readily available to assess the accuracy of IC systems at a single point for V̇CO2 and V̇O2, the validity of such procedures is limited when used in testing protocols that incorporate a wide range of energy production (e.g., basal metabolic rate and maximal exercise testing). Accordingly, we built an apparatus that allowed us to manipulate propane burn rates in such a way as to assess the linearity of IC systems. This technical report aimed to assess the accuracy and linearity of three IC systems using our in-house built validation procedure.

Approach

A series of trials at different propane burn rates (PBR) (i.e., 200, 300, 400, 500, and 600 mL min−1) were run on three IC systems: Sable, Moxus, and Oxycon Pro. The experimental values for V̇O2 and V̇CO2 measured on the three IC systems were compared to theoretical stoichiometry values.

Results

A linear relationship was observed between increasing PBR and measured values for V̇O2and V̇CO2 (99.6%, 99.2%, 94.8% for the Sable, Moxus, and Jaeger IC systems, respectively). In terms of system error, the Jaeger system had significantly (p < 0.001) greater V̇O2(mean difference (M) = −0.057, standard error (SE) = 0.004), and V̇CO2(M = −0.048, SE = 0.002) error compared to either the Sable (V̇O2, M = 0.044, SE = 0.004; V̇CO2, M = 0.024, SE = 0.002) or the Moxus (V̇O2, M = 0.046, SE = 0.004; V̇CO2, M = 0.025, SE = 0.002) IC systems. There were no significant differences between the Sable or Moxus IC systems.

Conclusion

The multiple PBR approach permitted the assessment of linearity of IC systems in addition to determining the accuracy of fractions of expired gases.

Introduction

Of primary interest to use indirect calorimetry (IC) in the study of human thermoregulation is the measurement of fractions of expired oxygen (FeO2) and carbon dioxide (FeCO2) for estimation of energy production or substrate turnover under various environmental conditions (e.g., at rest, during exercise, and in cold or hot environments) (Jequier & Felber, 1987). Chemical, electronic, and spectroscopic technologies have been implemented to perform these measurements. Indirect calorimetry systems integrate discrete electronic analyzers to record FeO2 and FeCO2 in line with measures of flow rate (e.g., flowthrough respirometry) or ventilation rate (e.g., breath-by-breath measurements) in addition to the temperature, pressure, and humidity of ambient and expired air using computer-controlled analog-to-digital signal processing (Leonard, 2012). Several instrument configurations range from simple or semi-automated mixing chamber systems to highly sophisticated, fully automated breath-by-breath measurement devices (Matarese, 1997). Each instrument incorporated in the IC system contributes errors in measuring whole-body oxygen uptake (V̇O2) and carbon dioxide production (V̇CO2), affecting estimates of energy production and substrate turnover. The expected accuracy of IC systems can differ between manufacturers, and the user is often left to rely on the devices’ technical notes to estimate the accuracy of their measured outcomes. However, technical notes generally report specifications for the individual analyzers and not the IC system as a whole. Accordingly, additional procedures are needed to assess the accuracy of IC measures under conditions that reflect the testing environment, especially when investigating small metabolic differences between the experimental and control conditions (Lighton, 2008).

Researchers and clinicians in the biological/medical sciences routinely implement IC technology to study the effects of behavioral and environmental manipulations on whole-body energy metabolism in humans under resting and exercise conditions (Brooks, Fahey & Baldwin, 2005). Several techniques have been developed to assess the accuracy of outcomes recorded through IC; however, few are designed to simulate the energy production rates recorded among humans at rest and during light physical activity. Burning of methanol (Cooper & Storer, 2001) or propane (Lighton, 2008) and nitrogen dilution (Fedak, Rome & Seeherman, 1981) are the most commonly applied techniques to validate IC measurements (i.e., V̇O2 and V̇CO2). Yet, the determination of signal linearity through the generation of multiple propane burn rates or levels of nitrogen dilution remains rarely performed. Lack of appropriate validation may cause inaccurate interpretations of V̇O2 and V̇CO2 measurements and, consequently, lead to large errors in calculating substrate partitioning and energy production (Ferrannini, 1988). Advancements in sensor technologies will continue to increase the accuracy and reliability of IC measurements in both laboratory and field settings and render energy production assessment promptly available (Haugen, Chan & Li, 2007). However, the accuracy and linearity of the measurements under various experimental and clinical conditions warrant special considerations for the enhancement of existing methodologies used to validate IC outcomes (Levine, Eberhardt & Jensen, 1999). Therefore, the present manuscript reports a new propane gas validation procedure designed to assess the accuracy and linearity of V̇O2 and V̇CO2 measurements taken at different propane burn rates (PBR).

Materials and Methods

The accuracy and linearity of three IC systems available in our exercise physiology laboratory (i.e., ViaSys Jaeger Oxycon Pro- now CareFusion, Hochberg, Germany; Moxus Modular Metabolic System, AEI technologies, IL, USA; and Sable Classic Line, Sable Systems International, Las Vegas, NV, USA) were assessed with an in-house built device for manipulating PBR and recovering the products of complete propane combustion (i.e.,  CO2 and H2O) into the mixing chambers of the IC systems. All IC systems were set for flow-through respirometry measurements (i.e., excurrent flow measurement or negative pressure system). During Jaeger and Moxus measurements, subsampled air was dried by passing through a twin-tube Nafion sample line or tube filled with magnesium perchlorate, respectively. Water vapor pressure was recorded during Sable measurements, and flow rate (FR), FeO2, and F eCO2 were corrected (see equations below).

Technical characteristics of the indirect calorimetric systems

The ViaSys Jaeger Oxycon Pro is a quasi-modular IC system consisting of a twin tube (Nafion sample line), a turbine volume transducer (flow range = 0–300 L min−1; accuracy = 2%), a subsample pump (flow rate ranging from 200 to 220 ml min−1), a 6 L mixing chamber, a fuel-cell oxygen sensor (accuracy = 0.05%; resolution = 0.01%; full range = 0–25%; time response = 0.08 s), a dual infrared carbon dioxide sensor (accuracy = 0.05%; resolution = 0.01%; full range = 0–15%; time response = 0.08 s).

The Moxus Modular Metabolic System, consists of a turbine for determination of ventilation volume [VMM-400 (flow range = 0–800 L min−1; accuracy = 1%)], a 4.2 L mixing chamber, a subsample pump (flow rate ranging from 10-500 mL min−1), a zirconia oxygen sensor [3A/I Oxygen analyzers (accuracy = 0.01%; resolution = 0.01%; full range = 0–100%; time response = 0.1 s)] and a dual infrared carbon dioxide sensor [CD-3A Carbon Dioxide (accuracy = 0.02%; resolution = 0.01%; full range = 0–15%; time response = 0.025 s)]. To investigate differences in oxygen sensor technologies (i.e., zirconia vs. paramagnetic), Moxus IC system validity measures were recorded with its gas analyzers (3A/I and CD-3A) integrated with Sable technologies for generating FR and subsample rate (described below).

The Sable Classic Line is a modular IC system consisting of a subsample pump (sub-sampler, SS4 –linearized mass flow meter ranging from 0–2,000 ml min−1), a water vapor analyzer (RH-300 –resolution = 0.001% and full range = 0–100% RH non-condensing), a dual infrared carbon dioxide sensor [CA-10 Carbon Dioxide (accuracy = 1%; resolution = 0.00001%; full range = 0–10%; time response = 0.5 s), a paramagnetic oxygen sensor [PA-10 Oxygen analyzers (accuracy = 0.1%; resolution = 0.0001%; full range = 0–100%; time response = 0.2 s)] and an air mass flow generator and controller (FK-500 –accuracy = 0.05 L min−1; full range = 50–500 L min−1).

Calibration of IC systems

Before data collection, oxygen and carbon dioxide analyzers were calibrated with medically certified calibration gases (1% CO2 and 100% N2 for Sable and 4% CO2 and 16% O2 for Moxus and Jaeger). In terms of FR, the Oxycon Pro’s turbine volume transducer was calibrated using the manufacturer’s built-in automated calibration procedures. Outcomes of the Sable’s air mass flow generator and controller were validated using the nitrogen dilution technique described by Fedak, Rome & Seeherman (1981). A series of N2 gas flows (1,000, 500, 250. 150 ml min−1) were randomly selected at two different ventilation rates (55 and 75 L min−1) and injected into the incurrent air through a canopy. The actual experimental values of N2 gas were compared to their theoretical value (Fedak, Rome & Seeherman, 1981).

Multiple propane burn rate device

The overarching aim of the current methodological study was to develop a procedure to assess the validity of V̇O2 and V̇CO2 measurements recorded at rest and during light exercise in human subjects. Given that propane gas combustion provides one of the best methods for simulating whole-body energy metabolism (Rising et al., 2015), a device was built to enable the complete combustion of propane at different mass flow rates (described as propane burn rates (PBR) to avoid confusion with the flow rate used during flow-through respirometry) and the recovery of V̇O2 and V̇CO2. The in-house built propane gas validation system consists of the following sequential connections. First, a tank of chemically pure (99%) propane gas (SPG-PROCHP6 –Air-Liquids Canada) with a two-stage Western Medical gas regulator (model M1-940-PG, Westlake, Ohio) and its gas hose is connected to a one-way Matheson mass-flow transducer (model 8141) that is subsequently connected to a Matheson mass-flow controller, model 8240 (East Rutherford, NJ) and to a Bunsen burner –vertical metal tube of 60 mm high and four mm inside diameter (ID). The burner is located in a 2.4 L glass canopy that flows into a 0.4 L glass tubing. The entire system is connected to IC systems by a 1.4-inch diameter hose (Fig. 1). The pressure in the gas line flowing out of the cylinder is maintained at ten psi. The regulator is fitted with a 14 inch MNPT brass needle valve and a high-pressure gas hose (four mm ID, seven mm outside diameter, OD) to prevent potential propane leaks. The flow rates were expressed in STP-corrected volumes from the analog output that varied from 0 to 5 volts on a 0% to 100% full scale (Lighton, 2008). The data collection is detailed in the next section, and its results are shown in the results section.

Figure 1 Schematic representation of the in-house built propane gas device utilized to assess the accuracy and linearity of indirect calorimetry systems.

The propane gas device encompasses a tank of chemically pure propane gas with a two-stage Western Medical gas regulator and its gas hose connected to a one-way gas mass-flow transducer subsequently connected to a gas mass-flow controller, and to a Bunsen burner. The burner is located into a 2.4 L glass canopy that flows into a 0.4 L glass tubing. From there, the entire system is connected to IC by a 1.4-inch diameter hose.

Data collection and reduction

A series of PBRs (200, 300, 400, 500, and 600 mL min−1) were selected and tested at two different FRs to assess each IC system’s accuracy and linearity. According to the Jaeger recommendations for resting metabolic rate, the flow rate was set at 20 and 40 L min−1 using the digital volume transducer described above. For the Sable IC system and Moxus gas analyzers, flow rates were set at 55 and 75 L min−1. Three 30-min trials per PBR and FR were randomly performed on each IC system. Also, 15-min of baseline (i.e., room air) was recorded before and after each PBR trial equating to 60-min measurement periods. All trials were conducted at one location and at the same time of the day. In addition, the FeO2, FeCO2, FR, barometric pressure (BP), water vapor pressure (WVP), chamber temperature (T∘C-Ch), and room temperature (T∘C-Rm) were recorded.

Propane gas stoichiometry equations

Pure propane (C3H8) is an odorless, colorless, flammable gas. Complete combustion of one mole of 100% C3H8 produces three moles of CO2 and 4 moles of H2O for every five moles of O2 consumed according to the stoichiometry reaction depicted in Eq. (1). (1) C3H8+5O2=>3CO2+4H2O.

Therefore, under standard pressure and temperature (STPD), 22.44 L of C3H8 would react with 112.2 L of O2 for the reaction to be completed to produce 67.2 L of CO2. At standard conditions, the molecular mass of 100% C3H8 is 44 g, and 1 g of propane would then require 2.55 L of O2 (i.e., 112.2 L of O2 to burn 44 g of propane) to produce 1.53 L CO2, that is, 67.2 L of CO2 results from the burning of 44 g of propane. It can then be deduced that optimal C3H8 combustion results in RER equal to 0.60 (Lighton, 2008).

The present technical report used a mass flow meter for an accurate PBR. The PBR was calculated using the following formula: (2) Mass flow rategmin−1=volume flow rate×propane density.

Flow-through respirometry equations

The measured fraction of expired gases (FeO2 and FeCO2) and FR recorded by the Sable IC system were first corrected for the effect of WVP using Dalton’s law of partial pressures. (3) F′eO2=FeO2×BP/BP−WVP

Where F’eO2 and FeO2 represent a fraction of expired air dry and moist oxygen. (4) F′eCO2=FeCO2×BP/BP−WVP

Where F’eCO2 and FeCO2 represent a fraction of expired air dry and moist carbon dioxide. (5) FR′=FR×BP−WVP/BP

where FR’ and FR represent dry and moist air FR, respectively.

To correct for any drift in the fraction of oxygen, the following equation developed by Lieberman et al. (2015) was computed: (6) FiO2ss=FiO2f+FiO2f−FiO2i×Tss−Ti/Tf−Ti

where FiO2i is the initial fractional amount of oxygen in the inspired air stream measured at equilibrium before each PBR (baseline pre-); FiO2f is the final fractional amount of oxygen in the inspired air stream measured at equilibrium after each PBR (baseline post-); Tss is the time into each PBR at steady state; Tf is the time when final inspired oxygen fraction is measured; Ti is the time when initial inspired oxygen fraction is measured.

The calculation of V̇O2, V̇CO2, and RER was performed using the following equations: (7) V ˙O2=FReFiO2−Fe′O2−FiO2Fe′CO2−FiCO2/1−FiO2

where FRe is expired flow rate; FiO2 stands for the fraction of inspired unscrubbed oxygen; Fe’O2 stands for expired dry oxygen; Fe’CO2 for expired dry carbon dioxide; and FiCO2 stands for the fraction of inspired unscrubbed carbon dioxide. (8) V ˙CO2=FReFe′CO2−FiCO2+FiCO2FiO2−Fe′O2/1+FiCO2

where acronyms stand as in Eq. (7) (9) RER=V ˙CO2/V ˙O2

where RER is the quotient of V̇CO2 over V̇O2.

The metabolic data (V̇O2, V̇CO2, and RER) were then truncated by 10 min (5 min at the beginning and end of each data collection period). An average value was reported for the remaining 20 min. To determine the accuracy and linearity of the three different IC systems, respirometry data were compared to the stoichiometry theoretical V̇O2, and V̇CO2 values under standard conditions for the five PBRs studied. Given that the same propane burn rates were used to validate the three IC systems, data are expressed as the mean difference between stoichiometry theoretical value and mean experimental values (i.e., M Δ=STV-MEV).

Statistical analysis

Statistical analyses were performed using SPSS, version 23 (SPSS Inc., Chicago, IL, USA). Unless otherwise specified, all values are reported as mean ± standard deviation, and an alpha level of 0.05 was used to indicate statistical significance. Tests for statistical assumptions were performed; homogeneity of variance was tested using Levene’s test, and normality was tested using the Kolmogorov–Smirnov test. First, descriptive statistics were conducted. Second, a series of one-way ANOVA was used to assess the effect of ventilation rates (55 L min−1, 75 L min−1 for Sable and Moxus, 20 L min−1, 40 L min−1 for Jaeger) on the fraction of gases. Third, a linear regression analysis was performed to examine the linearity between the volumes of V̇O2, and V̇CO2, with PBR for the three systems. Regression analysis was also performed to test the linearity between N2 flow rates with V̇O2 during validation of the Sable mass flow generator and controller. Fourth, Bland-Altman plots followed by linear regressions were created to evaluate the mean difference (error) between systems outputs and the stoichiometry theoretical V̇O2, V̇CO2, and RER values for all systems at all PBRs. Lastly, two-factor ANOVAs were conducted to evaluate the effects of the three IC systems and the five PBRs for V̇O2,V̇CO2, and RER error. A mixed models design was used with systems being a fixed effect and PBR a random effect. A corrected F-test was calculated (Neter et al., 1996) for the random factor as mixed models in SPSS incorrectly uses MS from the interaction as the error term (denominator). The correction is to use MS error as the denominator. SPSS correctly uses MS error in the F-test for the interaction, which is also considered random. SPSS correctly uses the MS from the interaction as the error term (denominator) for the fixed factor’s F-test. Effect sizes were calculated for F-test: Omega-squared (Ω2) was used for the fixed effect of the system, and rho (ρ) was used for the random effect of flow rate as was the interaction effect. In case of significant interactions, Tukey and Bonferroni post-hoc tests were applied.

Results

Exploratory and descriptive statistics

Descriptive statistics were performed on V̇O2, V̇CO2, and RER (Table 1). Levene’s test for testing homogeneity of variance (V̇O2, V̇CO2) for PBRs was significant. Homogeneity of variance between levels of PBR was violated for all outcome variables for all systems. However, F-statistics are robust when there are no equal variances (Field, 2015), especially in a balanced design. A test of normality (Kolmogorov–Smirnov) was performed within PBRs. The assumption of normality was not met for all data. Within overall PBRs (regardless of system), normality was only met for the 200 ml min−1 condition for both V̇O2 and V̇CO2. When normality was examined by system, assumptions of normality were met for both V̇O2 and V̇CO2 for every PBR.

Table 1 Descriptive statistics of the Sable, Moxus, and Jaeger metabolic carts including the SD, SE, CV of mean V̇O2 (ml min−1), V̇CO2 (ml min−1), and RER.

System	PBR	V̇O2 (ml min−1)	V̇CO2 (ml min−1)	RER	
		Mean	SD	SE	CV	Mean difference	Mean	SD	SE	CV	Mean difference	Mean	SD	SE	CV	Mean difference	
Sable	200	263	14.0	5.7	0.05	−48	158	7.9	3.2	0.05	−29	0.60	0.01	0.00	0.02	0.00	
	300	363	6.9	2.8	0.02	−40	215	4.5	1.8	0.02	−21	0.59	0.01	0.00	0.02	0.01	
	400	478	5.8	2.4	0.01	−48	284	5.0	2.0	0.02	−26	0.59	0.01	0.00	0.02	0.01	
	500	585	12.6	5.1	0.02	−47	346	6.4	2.6	0.02	−23	0.59	0.01	0.00	0.02	0.01	
	600	685	9.0	3.7	0.01	−39	409	4.9	2.0	0.01	−22	0.59	0.01	0.00	0.02	0.01	
Moxus	200	245	5.8	2.4	0.02	−30	148	3.7	1.5	0.03	−19	0.60	0.02	0.01	0.03	0.00	
	300	368	4.2	1.7	0.01	−45	217	3.8	1.6	0.02	−23	0.59	0.01	0.00	0.02	0.01	
	400	486	14.6	6.0	0.03	−56	288	8.8	3.6	0.03	−30	0.59	0.01	0.01	0.02	0.01	
	500	583	21.8	8.9	0.04	−45	346	11.5	4.7	0.03	−23	0.59	0.02	0.01	0.03	0.01	
	600	700	12.6	5.1	0.02	−54	418	6.0	2.5	0.01	−31	0.59	0.01	0.00	0.02	0.00	
Jaeger	200	187	19.4	8.0	0.1	28	114	8.2	3.3	0.07	15	0.61	0.05	0.02	0.08	−0.01	
	300	281	26.5	10.8	0.1	43	163	11.9	4.9	0.07	31	0.58	0.03	0.01	0.05	0.02	
	400	368	40.6	16.6	0.1	62	208	20.1	8.2	0.1	50	0.56	0.01	0.01	0.02	0.03	
	500	474	31.9	13.0	0.07	64	262	17.7	7.2	0.07	61	0.55	0.01	0.00	0.02	0.04	
	600	560	42.2	17.2	0.08	86	307	26.8	11.0	0.09	80	0.57	0.01	0.01	0.02	0.05	
Notes.

PBR propane gas burn rate

V̇O2 volume of oxygen

V̇CO2 volume of carbon dioxide

SD standard deviation

SE standard error

CV coefficient of variance

RER respiratory exchange ratio

Validation of Sable mass flow generator

Simple linear regression analyses were conducted to assess the linear relationship between N2 flow rates and actual experimental V̇O2 (mL min1) for the Sable system. Results showed a strong linear relationship between the actual experimental V̇O2and the N2 flow rates (Radj2 = 0.996; β = 0.998; 95% CI [0.174–0.183]; p < 0.001).

Effect of flow rate on respirometry outcomes

A series of 15 one-way ANOVAs [2 FR (55 L min−1, 75 L min−1 for Sable and Moxus, 20 L min−1, 40 L min−1, for Jaeger) X 5 PBRs (200, 300, 400, 500, 600 mL min−1)] revealed no significant effect of FR on V̇O2 and V̇CO2 outcomes for the Sable, Moxus, and Jaeger IC systems. Accordingly, data was pooled for each PBR (i.e., FR as a variable was ignored). The average of the six experimental trials per PBR is shown in Table 1.

Linear relationship of V̇O2 and V̇CO2 by PBR per system

A linear regression analysis was run to assess the linearity between the V̇O2, V̇CO2, and the PBR through the determination of linear regression equations (y = b0 + bx + ɛ) for each system. Table 2 shows that for the Sable system, PBR explains 99.6% of the variability in V̇O2 and V̇CO2 (R2 = 0.996), while it explains 99.2% and 99.4% of the variability in V̇O2 and V̇CO2, respectively, for the Moxus. However, the Jaeger system had the worst linear scores compared to the other two systems (94.8%, and 94.2%, for V̇O2 and V̇CO2, respectively). Furthermore, the mean values of V̇O2 and V̇CO2 were lowest for the 200 ml min−1 condition and increased as PBR increased for the three systems (Fig. 2).

Table 2 Linearity analysis of the Sable, Moxus, and Jaeger metabolic carts including Beta, R2, Confidence Interval (CI), and p-value of V̇O2 (ml min−1) and V̇CO2 (ml min−1) errors.

System	V̇O2 (ml min−1) Error	V̇CO2 (ml min−1) Error	
	β	R 2	95% CI	p-value	β	R 2	95% CI	p-value	
Sable	0.998	0.996	(0.001, 0.001)	0.001	0.998	0.996	(0.001,0.001)	0.001	
Moxus	0.996	0.992	(0.001, 0.001)	0.001	0.997	0.994	(0.001,0.001)	0.001	
Jaeger	0.974	0.948	(0.001, 0.001)	0.001	0.944	0.942	(0.00, 0.001)	0.001	
Notes.

V̇O2 volume of oxygen per minute

V̇CO2 volume of carbon dioxide per minute

Figure 2 Mean rates of oxygen uptake (VO2, L min−1.

(A) Carbon dioxide production (VCO2, L min−1; (B) and respiratory exchange ratio (RER; (C) measured during propane gas combustion tests with three metabolic carts: Sable system (dashed line), Moxus system (solid line), and Jaeger system (dotted line). Values are means SD.

Assessing error of V̇O2, V̇CO2, and RER for each system

To assess the relationship between the error in the system (V̇O2, V̇CO2, and RER) and the PBR, Bland-Altman plots were created and were followed by linear regression analysis for each system. As shown in Table 3, there were weak, moderate, and strong linear relationships between the V̇O2, V̇CO2 errors, and the PBR for the Sable, Moxus, and Jaeger systems, respectively. Also, V̇O2 and V̇CO2 errors of the Sable system had a non-significant p-value, which indicates that the Sable system had the lowest error in both volumes compared to the other two systems. However, although p-values were significant for both Moxus and Jaeger systems, the V̇O2 and V̇CO2 errors of the Moxus system had a weaker relationship with PBR and, therefore, lower error than the Jaeger system. The Jaeger system had the highest error among the three systems.

A two-factor [3 (IC) ×5 (PBR’s)] ANOVA was conducted to evaluate the effects of the three systems and the five PBR’s on V̇O2, V̇CO2, and RER error. For V̇O2 the results indicate a significant interaction between system and propane burn rate (F(8,75) = 3.328, p = 0.0026). Approximately 26.6% (ρ = 0.266) of the variance in V̇O2 error is accounted for by the interaction factor. The results show a non-significant main effect of propane burn rate on V̇O2error (F(4,75) = 1.0714, p = 0.3767), and it accounts for only 0.2% (ρ = 0.00286) of the variability in V̇O2 error. The main fixed effect of system was found to be significant (F(2,8) = 67.028, p < 0.001). All else held constant, the fixed main effect of system accounts for approximately 83% (Ω2 = 0.8293) of the variability in V̇O2. The Jaeger system had significantly (p < 0.001) greater V̇O2error (M = −0.057, SE = 0.004) compared to either the Sable (M = 0.044, SE = 0.004) or Moxus (M = 0.046, SE = 0.004) systems. There were no significant differences between Sable and Moxus systems on the above-mentioned variables (Fig. 3).

The results for V̇CO2 indicate a significant interaction between systems and PBR (F(8,75) = 10.722, p < 0.001). Approximately 54% (ρ = 0.539) of the variance in V̇CO2 is accounted for by the interaction factor. The results show a significant main effect of PBR on V̇CO2(F(4,75) = 9.375, p < 0.001) and it accounts for 14.8% (ρ = 0.148) of the variability in V̇CO2. The main fixed effect of system is also significant (F(2,8) = 34.966, p < 0.001). All else held constant, the fixed main effect of system accounts for approximately 89% (Ω2 = 0.8947) of the variability in V̇CO2. The Jaeger system had significantly (p < 0.001) greater V̇CO2 error (M = −0.048, SE = 0.002) compared to either the Sable (M = 0.024, SE = 0.002) or Moxus (M = 0.025, SE = 0.002) systems. There were no significant differences between Sable and Moxus systems on the above-mentioned variable (Fig. 3).

The results for RER indicate a significant interaction between system and PBR (F(8,75) = 3.332, p = 0.003). Approximately 20% (ρ = 0.2044) of the variance in RER is accounted for by the interaction factor. The results show a significant main effect of PBR on RER (F (4,75) = 7.8125, p < 0.001) and it accounts for 21.8% (ρ = 0.2184) of the variability in RER. The main fixed effect of system is also significant (F(2,8) = 4.993, p = 0.039). All else held constant, the fixed main effect of system accounts for approximately 23.8% (Ω2 = 0.2381) of the variability in RER. The Jaeger system had significantly (p < 0.001) greater RER error (M = −0.028, SE = 0.003) compared to either the Sable (M = −0.005, SE = 0.003) or Moxus (M = −0.005, SE = 0.003) systems. There were no significant differences between Sable and Moxus systems on the above-mentioned variable (Fig. 3).

Discussion

Laboratories using IC routinely implement propane gas validation techniques (Melanson et al., 2010; White et al., 1996). However, procedures are rarely performed to assess both the accuracy and the linearity of individual systems. We, therefore, built a device that enabled the manipulation of PBR during flow-through respirometry measurements. The custom-built device was used to evaluate the accuracy and linearity of IC systems readily available in our exercise physiology laboratory. Also, the multiple PBR procedure was implemented at two flow rates with each IC system (55 L min−1, 75 L min−1 for Sable and Moxus, 20 L min−1, 40 L min−1, for Jaeger) to investigate the effect of flow rate on V̇O2 and V̇CO2 measurements. The primary outcome of the current study revealed that the multiple PBR procedure agreed with the stoichiometric theoretical values (Table 2), and strong linear responses were observed, suggesting that the technique could be used to evaluate the accuracy and linearity of IC systems. Further, burning propane gas at rates within the range of whole-body energy metabolism in human subjects at rest and during light physical activity revealed differences in accuracy and linearity between IC systems.

Table 3 Regression analysis of V̇O2 (ml min-1), V̇CO2 (ml min-1), and RER errors by propane flow levels which followed the Bland-Altman plots for each metabolic cart (Sable, Moxus, Jaeger).

System	V̇O2 (ml min−1) Error	V̇CO2 (ml min−1) Error	RER Error	
	β	SE	R adj 2	t-value (p-value)	β	SE	R adj 2	t-value (p-value)	β	SE	R adj 2	t-value (p-value)	
Sable	−0.168	0.000	−0.006	−0.903 (0.37)	−0.281	0.000	0.046	−1.552 (0.13)	−0.379	0.000	0.134	−3.848 (0.001)	
Moxus	0.438	0.000	0.163	2.575(0.02)	0.425	0.000	0.151	2.484 (0.02)	−0.162	0.000	−0.009	0.869 (0.4)	
Jaeger	−0.536	0.000	0.262	−3.363 (0.002)	−0.807	0.000	0.639	−7.227 (0.001)	−0.679	0.000	0.441	−4.891 (0.001)	
Notes.

V̇O2 volume of oxygen per minute

V̇CO2 volume of carbon dioxide per minute

SE standard error

RER respiratory exchange ratio

Figure 3 Mean difference (delta) between the stoichiometric theoretical values and the actual experimental values of oxygen uptake (V̇O2, L min−1.

(A) Carbon dioxide production (V̇CO2, L min−1; (B) and respiratory exchange ratio (RER; C) measured during propane gas combustion tests with three indirect calorimetric systems: Sable system (dashed line), Moxus system (solid line), and Jaeger system (dotted line). Values are means.

Accuracy and linearity of the indirect calorimetry systems

The accuracy of the IC system typically refers to the closeness between the measured value and the “true” value (theoretical stoichiometry value). In contrast, linearity refers to systematic and random errors (Cooper & Storer, 2001). Regression analyses and ANOVA were performed to determine the accuracy and linearity of the three IC systems on V̇O2, V̇CO2, and RER. The ventilation rates were pooled because statistical outcomes showed that the two ventilation rates at the studied PBRs did not alter the V̇O2 and V̇CO2. As shown in Table 1, the mean difference between experimental and theoretical values of the Sable and Moxus systems overestimated V̇O2, and the mean differences were stable across the 5 PBRs. On the contrary, the Jaeger system underestimated V̇O2, and the mean differences increased with progressing PBR. The multiple PBR procedure demonstrated a high consistency for RER values for both Sable and Moxus systems. Our technical report showed that both Sable and Moxus systems responded similarly to propane gas combustion. However, it is important to note that the Moxus O2 and CO2 analyzers were incorporated into the Sable Classic Line technologies. Accordingly, further studies are needed using the complete Moxus Modular Metabolic System before making comparisons between the two IC systems. Special consideration should be given to the technologies used to measure FR during flow-through respirometry. The Sable system incorporates a mass flow meter, which is not affected by fluctuations in temperature and pressure (Lighton, 2008). However, systems that integrate volumetric flow meters need to correct for changes in temperature and pressure (Lighton, 2008). The increased discrepancy between experimental and theoretical values observed with the Jaeger system at increasing PBR may reflect a lack of appropriate correction for temperature changes.

The regression analysis of PBR vs. experimental values revealed a robust linear relationship for the three systems, meaning that both V̇O2 and V̇CO2 increased with propane gas combustion (Table 2). However, when linear regression was performed on the V̇O2 and V̇CO2 errors (difference), the Sable system provided the most accurate measurements evidenced by the least error values (i.e., Radj2 was the least with a non-significant p values (p = 0.37, p = 0.13) for V̇O2and V̇CO2, respectively).

The Sable and Moxus systems showed no change in mean difference for V̇O2, V̇CO2, and RER throughout progressive PBRs (Fig. 3), indicating more stable and reliable metabolic outcomes than the Jaeger system. The observed mean differences between analyzers could stem from the type of O2 analyzer (sensor) implemented in these systems. As time elapses, the fuel-cell oxygen sensor becomes noisy, unstable, and unreliable, an issue not always addressed by the manufacturer’s conventional medical gas calibration procedures. This is particularly true for fuel cells with faster response times but shorter lifespans. Conversely, paramagnetic (Sable System) and zirconia (Moxus) oxygen analyzers do not degrade over time if well maintained. Therefore, one cannot discard the temperature effect to account for the difference in O2 response. The paramagnetic oxygen analyzer has a temperature compensation circuitry that efficiently reduces the impact of temperature fluctuations. In contrast, the zirconia-cell oxygen analyzer does not fluctuate with temperature change owing to its high temperature operating capability. However, a sudden shift in temperature or a high operating temperature may induce significant drift in the fuel-cell O2 analyzer (Lighton, 2008), which the thermal compensation array may not correct. On the other hand, the CO2 analyzers offer far more stable readings with little to no drift. Accordingly, the observed discrepancy between experimental and theoretical values for both V̇O2 and V̇CO2 with increasing PBR (Fig. 3B) suggests that the effects of temperature on flow rate may explain the poor performance of the Jaeger system compared to the other two IC systems.

Reliability of the indirect calorimetry systems

Reliability refers to the reproducibility or repeatability of the readings of the gas analyzer under identical conditions (Cooper & Storer, 2001). The reliability of each IC system was determined by calculating the coefficient of variation (CV) of V̇O2 and V̇CO2. Our results showed that CV values of V̇O2 and V̇CO2 ranged from 1% to 5% and from 1% to 4% for the Sable and Moxus systems, respectively. On the other hand, the Jaeger system had CV ranging from 7% to 10% for V̇O2 and V̇CO2, respectively. Our data indicate that the Sable and Moxus systems provided more reliable measures of V̇O2 and V̇CO2 compared to the Jaeger system.

Propane gas technique compared to other techniques

The present study was conducted to validate the outcomes of IC systems using commercially available gas mass flow devices, and as such, it represents a novel approach in comparison to techniques/procedures routinely performed in other human physiology laboratories. For example, Melanson et al. (2010) conducted propane gas combustion tests to validate the outcomes of their whole room IC system, which integrated similar Sable System technologies (the O2 analyzer incorporated a fuel cell rather than paramagnetic). The Sable system provided more than 98% of the expected recovery of V̇O2 and V̇CO2 during the propane combustion trials. In addition to validation through propane combustion tests, the authors also investigated the system’s ability to detect known changes in energy metabolism in humans transitioning from rest to light exercise, and then to more moderate intensity exercise. Again, the whole room IC system performed as anticipated. However, in the latter experiment, the expected accuracy of the measurements taken at different metabolic rates is dependent on the systems linearity. Accordingly, performing the propane combustion test at multiple PBRs, as outlined in the current study, would strengthen the interpretation of these outcomes.

In a recent study, Rising et al. (2015) used propane gas combustion to determine the accuracy of another whole room Sable metabolic system. The IC system was subjected to multiple (n = 10) propane (99.5% purity) combustion tests to simulate 24-hour metabolic measurements. The burn rate (0.15 ± 0.025 g/min) was determined by obtaining the weight before and after completion of each combustion test using a calibrated analytical balance. Although within acceptable and non-statistically different limits, the Sable system was observed to under report mean V̇O2 values when comparing the outcomes of combustion to that of propane stoichiometry (Rising et al., 2015). In the current study, mean experimental values for V̇O2 were higher than the stoichiometry theoretical value at the five PBRs tested. Slight discrepancies between studies could be explained by differences in oxygen sensor technology (i.e., fuel cell vs. paramagnetic), indirect calorimetry method (i.e., ventilated hood vs. whole room) and validation procedure. The current multiple burn rate procedure adds to previously implemented propane combustion validation techniques by enabling the assessment of system accuracy and linearity within a single session. Furthermore, assessment of the accuracy and linearity of metabolic outcomes could allow for the correction of systematic error, which will be the focus of future work.

Other commonly used techniques to calibrate IC systems include alcohol combustion and butane gas burning. The two methods bear on stoichiometry calculations similar to that for the combustion of propane gas (i.e., O2 is consumed, and CO2 and water are produced) (Toien, 2013). For instance, Marks et al. (1987) calculated V̇O2 and V̇CO2 after burning known masses of ethanol and methanol gases (pure alcohol) in an open-circuit IC system. The authors reported that gas volumes differed by less than 5% from their theoretical values, confirming alcohol combustion as a valid technique for calibrating IC systems in neonates. Furthermore, Miodownik et al. (1998) described a methanol-burning lung model, which reported less than 0.005% reading error attributed to carbon monoxide gas production. Finally, Nunn, Makita & Royston (1989) showed that a known amount of butane gas burned in a closed-circuit IC system yielded, for one mole of butane, 6.5 moles of O2, and an RER of 0.61, providing a simple and robust technique to test IC system performance. However, although all these procedures produced good outcomes, they are not routinely implemented in a manner to evaluate the linearity of IC systems. quickly and efficiently. Therefore, the use of gas mass flow meter technology to monitor PBR represents a more comprehensive procedure that will enhance the validation IC outcomes under a variety experimental condition. In fact, Perez-Suarez et al. (2018) published a quality control study in which they implemented the proposed PBR method based on our approach (Ismail, 2017).

Conclusion

This work aimed to refine the propane gas validation technique. We had no intent to compare indirect calorimetry systems to promote one over the others. Each system was built for specific applications and, as such, differed from each other. The current technical report describes a complimentary propane gas validation procedure to assess the accuracy and linearity of indirect calorimetry. The IC systems output must yield an accurate fraction of gases for the computation of V̇O2 and V̇CO2 and energy production in medicine, nutrition, and exercise sciences. Generally, companies manufacturing indirect calorimetry systems provide customers with calibration procedures to address accuracy issues. However, to ascertain the quality of data acquisition, IC’s accuracy and linear response should be validated regularly. We recommend using the described multiple propane burn rate validation procedure to assess the accuracy and linearity of IC systems.

Additional Information and Declarations

Competing Interests

Author Contributions

Data Availability

The authors declare there are no competing interests.

Mohammad Ismail conceived and designed the experiments, performed the experiments, analyzed the data, prepared figures and/or tables, authored or reviewed drafts of the article, and approved the final draft.

Sanaa A. Alsubheen analyzed the data, authored or reviewed drafts of the article, and approved the final draft.

Angela Loucks-Atkinson analyzed the data, prepared figures and/or tables, authored or reviewed drafts of the article, and approved the final draft.

Matthew Atkinson analyzed the data, prepared figures and/or tables, authored or reviewed drafts of the article, and approved the final draft.

Thamir Alkanani conceived and designed the experiments, performed the experiments, analyzed the data, authored or reviewed drafts of the article, and approved the final draft.

Liam P. Kelly conceived and designed the experiments, authored or reviewed drafts of the article, and approved the final draft.

Fabien Basset conceived and designed the experiments, performed the experiments, analyzed the data, authored or reviewed drafts of the article, supervision of Mr. Mohammad Ismail Master’s thesis, and approved the final draft.

The following information was supplied regarding data availability:

Data is available at Borealis Dataverse:

Basset, Fabien, 2022, “Multiple propane gas burn rates method to determine accuracy and linearity of indirect calorimetry systems: an experimental assessment of a method”, https://doi.org/10.5683/SP3/9WYIXK, Borealis, V1, UNF:6:tfdn59vBL7Lux4ynNShlRQ== [fileUNF]

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
