# Peer review of "Multiple propane gas burn rates procedure to determine accuracy and linearity of indirect calorimetry systems: an experimental assessment of a method"

_PeerJ, doi:10.7717/peerj.13882_

## Round 0.1 · original submission · Major Revisions

Please address the comments provided by the reviewers carefully.

Reviewer 1 ·

Basic reporting

The basic reporting of the study is good. The English is clear and acceptable; some copyediting will be required because not all of the authors have English as their first language. I had no difficulty accessing the supplementary material, and it was clearly presented. One small caveat; because gas concentrations are expressed as fractional concentrations, the number of displayed decimal places is often far too small. This is not a huge problem, because the underlying data are present at full resolution and can be selected for display if desired. Hopefully all likely users of the result spreadsheets will be aware of this.

The literature survey was adequate. Line numbers would have been helpful for reviewers.

The authors do a good job of describing why the study was carried out. It is indeed the case that the vast majority of propane and other recovery measurements are only undertaken at one point, and the linearity of the indirect calorimetry system is hardly ever tested. This is especially significant because the linearity of many CO2 analyzers may be problematic. So it is an excellent point that the authors address.

Experimental design

I have no issues with regard to the experimental design.

With regard to equation 2, the authors may wish to mention that water vapor dilution is corrected by using an implementation of Dalton’s law of partial pressures. The vast majority of investigators try to remove water vapor from the air stream (as in the Moxus and Jaeger systems) rather than correcting for its presence, so an explanation might be a good idea. The idea of mathematically correcting for water vapor dilution is still a foreign concept to most researchers.

The statistical methods described seem appropriate. I’m impressed that the authors are using Bland-Altman plots, which are not widely known but are ideal for this kind of analysis.

Validity of the findings

I have no issues with the outcomes of the statistical tests described by the authors.

I was frankly a bit surprised by the poor performance of the Jaeger system, which appeared to display poor linearity for both O2 and CO2. The fact that the Delta RER was also a function of propane flow in the latter system suggests that the linearity of the CO2 analyzer is particularly suspect (fig. 3).

The authors state that Rising et al. (2015) found that a Sable system underestimated VO2 while overestimating VCO2. I am familiar with that MS and these conclusions are not supported by the MS in question. Perhaps a misreading by the authors?

Additional comments

At the beginning of materials and methods, the first IC system is “Sable Systems International”, not “Sables Systems International”. Sable is frequently misspelled as “Sables” elsewhere in the MS.

Equation 2 & equation 3 are both missing the divide sign before the denominator, as are equations 4, 5, 6, 7 and 8. For example, the right hand term for equation 2 should read FO2 x BP/(BP - WVP), not FO2 x BP (BP - WVP). The/symbol appears to be uniformly lacking in the equations.

The authors should double-check equation 7. I believe the denominator should be (1+ FiCO2). To be sure, this minuscule difference will not have a significant effect on the results of the paper.

For figure one, I suggest that a photograph of the actual combustion chamber be added to the supplementary material.

·

Basic reporting

This manuscript presents findings regarding a new multi-flow propane validation technique for three different metabolic carts (Sable, Moxus and Jaeger).
There are several problems in regards to how parts of the manuscript are written. First, it should be made clear in the beginning that the indirect calorimeter systems they are utilizing for their study are metabolic carts. They should also change the title to reflect this. Secondly, it is not clear if all three metabolic carts utilized ventilated hoods or mouth pieces’/face masks for metabolic measurements in human subjects. Indirect calorimetry systems can be anything from metabolic carts, that measure oxygen only, to a full blown whole room indirect calorimeters connected to research grade instrumentation that measures both oxygen and carbon dioxide concentrations. The major difference between the two is that metabolic carts require a physical connection between the subject and instrumentation verses that for a whole room indirect calorimeter.

Experimental design

The experimental design is fine if used only for metabolic carts. However, there is no mention about some of the inherent problems when burning ethanol or methanol utilizing burn kits. An extensive review of all the issues in relation to each gas or liquid utilized for validating all types of indirect calorimeters was recently published (Rising R, Foerster T, Avigdor DA, Albu J, Pi_Sunyer X. Validation of whole room indirect calorimeters: refinement of current methodologies. Physiol Rep. 2017; 5:22.). As mentioned in the reference alcohols do not oxidize completely thus the reason a smell can be detected during a burn. The recommended burn kits are the most standardized way commercially available to validate metabolic carts to date, despite its misgivings. The authors should have included some of this information in the Methods section of the manuscript.
The 15-minute test length of the propane combustion tests is not a standard duration for human subject metabolic measurements utilizing metabolic carts. The authors should have run their tests for 45-minutes, which would have included the 15-minute baseline and 30 minutes of actual metabolic measurement. A minimum of thirty-minutes is considered the standard length for human subject metabolic measurements utilizing metabolic carts (Schadewaldt P, Nowotny B, Strabburger K, Kotzka J, Roden M. Indirect calorimetry in humans: a postcalorimetric evaluation procedure for correction of metabolic monitor variability. Am J Clin Nutr. 2013;97: 763-73). Furthermore, the authors do not mention how the propane gas was introduced to each of the metabolic carts tested. It was not clear whether it was introduced directly to each metabolic carts sensors or was it just released under the ventilated hood which would have emulated a subject metabolic test. This information should be provided in the Methods section of the manuscript.
Presentation of the statistical analysis was well presented along with related tables and Figures.

Validity of the findings

The metabolic carts they were evaluating had oxygen and carbon sensors with different measurement ranges and technologies. The authors do mention some of the limitations of their findings in this regard. Furthermore, it was a good idea to test each of the metabolic carts at different ventilation rates. However, they do not mention that when metabolic carts are in use for subject metabolic measurements, the constant changes in air flow rate necessary to maintain carbon dioxide below 1% may introduce errors in the overall metabolic calculations. The authors should review the cited manuscript by Rising et al, 2015 for a good explanation of this. Some of this information should have been included in the Discussion section of the manuscript.

Additional comments

One problem is that their new calibration system is a home engineered device. Therefore, how would other research institutions be able to either 1) purchase the device or 2) be able to create their own? There is no mention of a future to commercialize it or provide specs to other institutions. Some mention of a future I feel would give this manuscript more meaning to readers. How would others incorporate this technique? As much as ethanol kits that come with metabolic carts are not the most accurate way to validate these instruments (Rising et al, 2015), they are commercially available from just about any metabolic cart manufacturer that recommends them. Knowing these issue, this reviewer actually created a simple system to validate metabolic carts with ventilated hoods utilizing propane combustion. However, it was a static type device with just one propane flow setting (unpublished observations). Finally, the one major advantage of traditional propane combustion tests is that only an analytical balance (available from multiple vendors) and lecture bottles of pure (99.5 % purity) propane gas (available worldwide from many vendors) are required. This, along with methanol burn kits, allows just about any institution conducting metabolic testing in human subjects with indirect calorimetry to validate their instruments. The only real advantage of the author’s device is the ability to determine the linearity of a metabolic carts sensors. However, this could also be done by direct injection of calibration gases with different concentrations of oxygen and carbon dioxide utilizing a precision gas blender (Rising et al, 2017). Moreover, this device is not suitable for whole room indirect calorimeters. The authors of this manuscript should mention some of these limitations in the Discussion section.
The information being provided and referenced (Rising et al, 2015) is incorrect. First of all, the data the authors are referring to are from one-hour simulated metabolic tests with both the metabolic cart (ethanol burns) and whole room indirect calorimeter (propane combustion) in question. The results for both techniques were extrapolated to 24-hours. The whole room calorimeter extrapolated simulated 24-hour resting metabolic rate, ventilation rates of both oxygen (VO2) and carbon dioxide (VCO2) were in error by 2.6, 1.4 and 1.0 %, respectively, in comparison to propane stoichiometry. The same error values obtained from the metabolic cart were 17.1, 11.1 and 11.8 %, respectively (Rising et al, 2015). Furthermore, if the authors read the method section of this reference, they would realize that the metabolic data from the whole room indirect calorimeter that was connected to the Sable instrumentation were derived after correction for the volume of the room via a z-transformation (Bartholomew GA, Vleck D, Vleck CM. Instantaneous measurements of oxygen consumption during pre-flight warm-up and post-flight cooling in sphingid and saturniid moths. J Exp Biol. 1981;90: 17-32). Since the actual oxygen and carbon dioxide readings are quite low during the one-hour propane combustion test, the metabolic data depend greatly on the z-transformation due to the high time constants. Despite this the errors are quite a bit lower for the whole room indirect calorimeter verses that of the metabolic cart that does not rely on such a calculation. Finally, the ultimate test of any validation procedure is how the instrument performs in regard to metabolic testing in human subjects. In a previous study (Hall KD, Chen KY, Guo J, Lam YY, Leibel RL, Mayer LE, Reitman ML, Rosenbaum M, Smith SR, Walsh BT, Ravussin E. Energy expenditure and body composition changes after an isocaloric ketogenic diet in overweight and obese men. Am J Clin Nutr. 2016;104:24-33) cited by Rising et al, 2017, whereby human subjects that resided on a metabolic ward for three months, had nearly perfect weight maintenance during this time. Their energy requirements were derived by a whole room indirect calorimeter previously validated by propane combustion (Rising et al, 2017).
The author of the reference in question (Rising et al, 2015), and a reviewer of this manuscript, has 40 years’ experience in conducting propane and ethanol combustion tests in regards to validating whole room indirect calorimeters and metabolic carts. Furthermore, this reviewer has created and validated whole room indirect calorimeter laboratories worldwide (Nas A, Mirza N, Hagele F, Kahlhofer J, Karschin J, Rising R, Kufer TA, Westphal AB.
Impact of breakfast skipping vs. dinner skipping on regulation of energy balance and metabolic risk. Am J Clin Nutr. 2017; 105:1351-1361). Finally, this author has published a manuscript (Rising et al, 2017) regarding standardized techniques in validating whole room indirect calorimeters utilizing a new shorter duration propane combustion test. In contrast to what the authors of this manuscript are suggesting, the pre and post weighing method of conducting propane combustion tests produces minimal error in the overall results. A hypothetical example is presented below:

a) Propane combustion test conducted for just one-hour.

b) Data extrapolated to 24-hours (per minute average x 1440 minutes in 24-hours).

c) Hypothetical results:

Total propane burned for one-hour = 10.00 g (Value read from balance correctly)
Total propane burned for one-hour = 10.02 g (Value read from balance 0.02 g high)
Total propane burned for one-hour = 09.98 g (Value read from balance 0.02 g low)

Titles in the table are in the following order: Parameters per 24-hours, True value (read correctly), Value (read low), Value (read high), % Error (true vs low), % Error (true vs high).

Parameters (True) (Low) (High) (% error low) (% error high)
Burn rate (g/min) 0.1667 0.1663 0.1670 0.24 0.18
Energy (kcal) 2861.4 2854.5 2866.5 0.24 0.18
VO2 (liters) 611.03 609.57 612.13 0.24 0.18
VCO2 (liters) 366.62 365.74 367.28 0.24 0.18

As can be seen from this hypothetical example, even if the balance is read incorrectly by 0.02 g, the errors in stoichiometry are minuscule as shown in the above table. This would mean, for example, that the 2.1% error for energy expenditure reported by Rising et al, 2015 for the whole room indirect calorimeter connected to the Sable instrumentation would be either 2.34% (low balance reading) or 1.92% (high balance reading). This is very close to the assumed 2% general error in metabolic results via indirect calorimetry due to non-oxidative processes within the human body. This is assuming this reviewer actually makes these kinds of errors in reading the analytical balance pre and post propane combustion. Finally, with such a low burn rate the balance is stable during the less than one second it takes to read and record the weight. During our propane combustion tests, the Sable instrument data acquisition is started and stopped and the balance read simultaneously in less than a second due to their close proximity. Therefore, errors even of 0.02 g high or low are highly unlikely. Further proof of this is presented by Rising et al, 2017. Therefore, this reviewer requests that this miss quoted information, either be removed or revised to reflect proper interpretation of the data. The differences found for the metabolic cart tested by Rising et al, 2015 are quite high in comparison to the whole room indirect calorimeter. Just the extrapolation calculations alone will magnify any potential errors for any whole room indirect calorimeter or metabolic cart, based on a one-hour propane or ethanol combustion validation. As seen by Rising et al, 2015, these can be quite high for metabolic carts.

Minor comments

a) The model numbers of each metabolic cart should be provided in the information
regarding the instrumentation.

b) There is no mention of the methodology utilized to calibrate and verify their mass flow
meter. It is mentioned in the Discussion section that it was calibrated but this
methodology is not presented. The authors should clearly state how the flow rates of
the mass flow meter was validated.

·

Basic reporting

* English language could be improved in parts. Some sections may be hard to comprehend for non-experts due to language.

* I recommend a pass through the Material and Methods section for consistency of the reporting. Ex. "... propane gas (Air-Liquids, Westlake, Ohio)" vs. "... a Matheson mass-flow transducer (model 8141)..."; "...(4 mm ID, 7mm outside diameter, OD)...". Likewise, there are a few instances of quantities reported without units. E.g. in Results/Exploratory Statistics "...was only met for the 200[ml/min] flowrate..."

* I also recommend to augment the accuracy figures given with a reference base (absolute; %reading; %full scale). As currently written some figures may be misleading for unsuspecting readers in direct comparison (e.g. CO2 accuracy Sable 1%, Moxus 1%, Jaeger 0.05%!!)

* In "Propane gas data collection" it should be made clearer how the three set-ups differed. As written, I get the impression that the Sable FK-500 was used to generate the main flow for all three systems. However, both ventilation rates (20/40 LPM) for the Jaeger cart were lower than the lower limit of the FK-500 (50-500LPM).

* There are numerous typographical errors in equations (?, it seems all slashes are missing):

eq2: F'iO2 = FiO2 x BP / (BP -WVP)
eq3: F'iCO2 = FiCO2 x BP / (BP -WVP)
eq4: FR' = FR x (BP -WVP) / BP
eq5: {[FiO2i + ((FiO2f - FiO2i) x Tss / (Tf - Ti))] - FeO2ss} x FR
eq6: VdotO2 = FRe x (Fe'O2 - FiO2 x Fe'CO2) / (1 - FiO2)
eq7: VdotCO2 = FRe x (Fe'CO2 - FiCO2 x Fe'O2) / (1 - FiCO2)
eq8: RER = VdotCO2 / VdotO2

* eq6 and eq7 as written seem incorrect. They apparently mix dry and wet concentrations and flow rates (if written in Lighton 2008 notation which was given as the initial reference of the section). eq 7 also uses (1-FiCO2) as a denominator which seems highly suspicious. Both equations 6 and 7 seem to miss an additional FiO2 and FiCO2 term in the numerator, respectively. Note that some equations in Lighton 2008 have been found to contain typographical errors and have been corrected in the book errata. If the authors deem the equations correct as written/cited, an explicit derivation or more specific reference (i.e. eq X in Lighton 2008) should be given.

* eq 5 should explicitly name the calculated quantity. As written it seems to correspond to (FiO2ss - FeO2ss) x FR, a simplified equation for VdotO2 that misses the Haldane correction necessary for compressible flows. The running text above it states that this was calculated to correct for O2 drift, which implies that a fractional O2 concentration was the intended quantity. I suggest to change eq5 to "FiO2ss = FiO2i + ((FiO2f - FiO2i) x (Tss - Ti ) / (Tf - Ti))" which implements a linear interpolation of FiO2 at time ss from a pre and post baseline measurement (FiO2i and FiO2f, resp.)

* Table 1: change definition of Mdelta to MEV-STV for consistency with the main text (see recovery comment in 2. Experimental Design). It also avoids the confusion that an overestimating cart has negative Mdelta. Also unify language in legend ("propane flow level" vs. "flow rate" in main text)

* Figure 1: indicate flow directions, indicate location of IC cart
* Figure 2: Add (thinner) lines for the theoretically expected values, VO2, VCO2, RER=0.6. Also a typo in the legend Jeager => Jaeger
* Figure 3: Same typo Jeager => Jaeger

Experimental design

* The difference between the concepts of "calibration" and "validation" should be made more explicit in the introduction. Overall, the authors perform a *validation* study using a novel technique for the accuracy and linearity of three IC devices. Yet, the introduction ends with the statement "...reports a new propane gas *calibration* procedure..."

* The author's custom built propane burning device seemingly was validated using an N2 dilution technique with one of the test devices used in the study (Sable). This results in a circular validation chain when using the propane burn device later to assess the linearity of the Sable device. This could partially explain good linearity results of the Sable IC cart, IF the N2 dilution data was used to calibrate the propane mass flow controller. The authors should clarify this point.

* In Results/Difference between ventilation rates... , errors are calculated as simple differences between measured and theoretical values. It is more common to express validation errors as relative errors or "recovery" which is normalized to expected signal amplitude:

recovery = (x_measured - x_theoretical) / x_theoretical = x_measured / x _theoretical - 1

Some justifying reasoning for the author's choice of absolute errors should be provided.

Validity of the findings

* Discussion on accuracy (last paragraph): "However, VCO2 is impacted by air temperature, a factor ...., but not well compensated with Jaeger .... significant VCO2 drift over time (Fig 3-B)". Please provide a reference for the VCO2 - air temperature link. Also Fig 3-B shows dCO2 ~ flow rate, i.e. does not illustrate CO2 drift over time.

* Dicussion/Propane gas technique: Note that both alternative Sable Systems IC systems referenced (Melanson 2010, Rising 2015) use different technology than the authors'. Both are fuel cell based systems, with Rising's 2015 system using automated calibration procedures for O2 spanning and WVP calibration. Inasmuch, direct comparisons on paper may be comparing the proverbial apples and oranges.

Also note that Melanson, 2010 uses constant monitoring of the propane gas tank weight via a scale with digital read out. Any fluctuation or decline in burn rate over time will be detected as changes in mass loss per unit time. The authors have a point that pre/post weight procedures generally are less accurate than the proposed method due to having to make the assumption of a constant, average burn rate whereas the authors' system actively controls burn rate to a constant propane flow. The limitation is not due to a "time lag" between measurements, however, nor will it "impact the ability to calculate sensor response time" (purely a function of room size, flow rate, sub sample flow rate and inherent sensor response time).

* Discussion: The discussion of techniques, in my opinion, would also greatly benefit from a comparison with more modern gas injection techniques based on CO2 augmented N2 dilution ("gas blending"). These have the same benefits as the authors' method over standard propane burns. However, through varying amounts of N2 and CO2 individually gas blending additionally allows to vary the RER of the input signal. In theory, both methods (controlled propane burn and gas blending) allow dynamic protocols with time varying input signal to simulate e.g. exercise protocols. Propane burns are claimed to have benefits for systems that do not measure water vapor pressure, due to the generation and injection of water vapor from the combustion, which simulates a subject's water loss. Gas injection techniques have benefits when it comes to fire safety for long term validations.

* Note that the lowest propane flow rate used in this study corresponds to ~290 ml/min of O2 consumption. This corresponds to a large human at rest. While there may have been technical limitations limiting the minimum propane flow rate, this means that a large fraction of the physiologically relevant signal range for an IC metabolic cart has not been explored in this study.

Additional comments

The authors provide a solid, although somewhat mechanical, validation study for 3 indirect calorimetry (IC) systems, specifically their linearity with varying flow rates and input signal strength.

---

## Round 0.2 · accepted · Accept

The reviewer supports the publication of this manuscript.

Reviewer 1 ·

Basic reporting

After a careful review of the referees' concerns & the authors' responses to them, I do not believe that any further changes are needed.

Experimental design

No issues here

Validity of the findings

I believe the findings are accurate & supported by evidence. After a careful review of the referees' concerns & the authors' responses to them, I do not believe that any further changes are needed.

Additional comments

Good revision. I have no further comments.